# *Cryphalus eriobotryae* sp. nov. (Coleoptera: Curculionidae: Scolytinae), a New Insect Pest of Loquat *Eriobotrya japonica* in China

**DOI:** 10.3390/insects10060180

**Published:** 2019-06-22

**Authors:** Sizhu Zheng, Andrew J. Johnson, You Li, Chunrong Chu, Jiri Hulcr

**Affiliations:** 1Suzhou Custom District, Suzhou 215000, China; zhengsizhu@126.com; 2School of Forest Resources and Conservation, University of Florida, Gainesville, FL 32611, USA; andrewjavanjohnson@gmail.com (A.J.J.); hulcr@ufl.edu (J.H.); 3Extension Center for Evergreen Fruit of Jiangsu Taihu, Suzhou 215107, China; peinzy@163.com

**Keywords:** Suzhou, taxonomy, canker, trunk, Asia

## Abstract

A previously unknown bark beetle species, *Cryphalus eriobotryae* sp. nov. Johnson, 2019 has emerged as a lethal pest of loquat (*Eriobotrya japonica*) in China. The description of new species has been provided. The new species is distinguished from the other *Cryphalus* by the weakly aciculate frons, by the antennae, with unevenly spaced procurved sutures, by the short pronotal disc, with hair-like setae, and by the widely spaced mesocoxae. The survey of plantation records from around Suzhou suggests that this beetle was introduced from another area not long before 2017. In the surveyed loquat plantation in 2018, 20–90% of trees showed signs of infestation, and 5% were killed in 2018, resulting in the death of over 1000 trees. Outbreaks of the apparently loquat-specific *Cryphalus eriobotryae* can be diagnosed by hundreds of cankers on the trunk, and wilted foliage. This pest is of concern as a loquat plantation pest and as a pest of fruit production and ornamental trees within Suzhou, and globally.

## 1. Introduction

Loquat *Eriobotrya japonica* (Thunb.) Lindl., 1822 (Rosaceae) is a fruit tree species native to Asia. The tree is grown worldwide as an ornamental tree with edible fruit, widely planted in Africa, Europe, the Mediterranean region and the Americas [1,2]. Besides its fruit, loquat is also prized as a landscape tree in many countries because of its lush evergreen foliage, and its distinctive environmental adaptability. In its native Asia, loquat is an important nectar source [3] for apiculture and is widely used by other insects [4].

Today an estimated 130,000 ha of loquat plantations exist in China with an annual production of over 650,000 tons. The commercial plantations of loquat in China account for more than 80% of the world’s population [5]. In 2016, one of the famous loquat planting provinces, the Jiangsu province, had a total of 2187 ha of planted loquat. Loquat products in this province are worth an estimated $32 million (USD). The number of plantations has expanded rapidly in recent years owing to a variety of factors including high market demand. Loquat plantations in Suzhou are about 1636 ha, which accounts for 75% of the Jiangsu province total loquat area [6].

There are few pests and diseases of the trunk reported in loquat [7]. The only notable fungal disease being black rot, caused by *Botryodiplodia malorum* (Berk.) Petr. & Syd., 1926 (Botryosphaeriaceae), which results in bark lesions [8]. Among insect pests, the longhorn beetles (Cerambicidae) *Apriona germai* Hope, 1831 and *Anoplophora chinensis* (Forster, 1771) may cause minor damage [9], but they are generalists, and they are not reported to be as lethal to loquat.

In April 2018, some loquat trees were attacked and killed by an unknown bark beetle in the commercial plantation at Suzhou, Jiangsu, China. It was considered as incidental natural death at first, but more trees nearby were killed by the same pest from August to October.

Bark beetles (Curculionidae: Scolytinae) are common in subtropical and temperate zones. They usually attack damaged or dying trees but can become a significant problem in monocultures or monodominant forest plantations [10]. For example, *Hypocryphalus dilutus* (Eichhoff, 1878) is a serious worldwide pest of mango and fig [11], and *Scolytus amygdali* Geurin-Meneville, 1847 is responsible for a significant loss of fruit production in almond orchards throughout the world [12]. To our knowledge, no bark beetle has been reported infesting and causing dieback of healthy loquat in Asia. This is the first report of loquat trees being killed by bark beetle attacks. In this article we describe a new species, *Cryphalus eriobotryae* Johnson (Curculionidae: Scolytinae: Cryphalini) and report the symptoms and damages to loquat, especially to loquat trees in plantations in Dongshan Town, Suzhou, Jiangsu Province, located in eastern China.

## 2. Materials and Methods

### 2.1. Taxonomy

The specimens were identified as *Cryphalus* based on Wood’s (1986) key to genera of Scolytinae and confirmed by comparison to specimens in the collection of one of the authors (AJJ). Specimens were examined using a stereomicroscope (Olympus SZX16) and photographed with a DSLR camera (Canon Rebel T3i) mounted on a compound microscope (Olympus BX53 with 5×, 10×, and 20× fluorite objectives). Montage images were assembled using a Helicon Focus Pro 7.0 (Helicon Soft, Kharkov, Ukraine).

The following entomological collection abbreviations are referenced in the text:NZMC-IZCAS—China, Beijing, Chinese Academy of Sciences, National Zoological Museum of China, Institute of Zoology.NIAES—Japan, Ibaraki, Tsukuba, National Institute of Agro-Environmental Sciences (ITLJ).NMNS—Taiwan, Taichung, National Museum of Natural Science.ZIN—Russia, St. Petersburg, Russian Academy of Sciences, Zoological Institute.BMNH—United Kingdom, London, The Natural History Museum.USNM—USA, Washington D.C., National Museum of Natural History.UFFE—USA, Florida, Gainesville, University of Florida Forest Entomology Collection.FSCA—USA, Florida, Gainesville, Florida State Collection of Arthropods.RIFID—South Korea, Namyangju, Research Institute of Forest Insect Diversity.

### 2.2. DNA Sequences

DNA was extracted from two specimens taken from the same collection material as the type series. DNA was extracted using a Qiagen DNeasy blood and tissue kit (Hilden, Germany). We amplified and sequenced the partial cytochrome oxidase I (COI) and the nuclear large ribosomal subunit (28S) using the primers used in Johnson et al. [11]. Reads were assembled using Geneious^®^ 9.1.8 (www.geneious.com). Sequences were used to search the databases GenBank (www.ncbi.nlm.nih.gov/genbank/) and BOLD (boldsystems.org) to identify potentially similar species.

### 2.3. Damage Surveys

Beginning in 2018, large-scale operational surveys of pests were conducted by the Extension Center for Evergreen Fruit of Jiangsu Taihu in Suzhou. All plantations in Dongshan Town were surveyed. Infestation and tree death were confirmed by visual inspection. Mortality of trees in nurseries was estimated from our survey and from reports by nursery owners and researchers from the Extension Center for Evergreen Fruit of Jiangsu Taihu.

## 3. Results and Discussion

Cryphalus eriobotryae Johnson sp. nov. 


Figure 1



www.zoobank.org/B7735C47-05CB-4107-B6E9-B56955314193


### 3.1. Material Examined

Holotype. (Female) Labelled “CHINA: Jiangsu. Suzhou.; Dongshan. 2018-IX-27; ex. *Eriobotrya japonica*; Chunrong CHU coll. IOZ(E)227908”( NZMC-IZCAS).

Paratypes. Same label information as holotype deposited in NZMC-IZCAS (2 males, 1 female); NIAES (1 male, 1 female), NMNS (1 male, 1 female), ZIN (1 male, 1 female), BMNH (1 male, 1 female), USNM (1 male, 1 female) UFFE (1 male, 1 female), FSCA (1 male, 1 female), RIFID (1 male, 1 female).

### 3.2. Diagnosis

*Cryphalus eriobotryae* can be diagnosed from other *Cryphalus* by the pronotum widest at the base, by the antennal club which has an unusually large area between the last apex and the last suture, by the male frons which is weakly aciculate and lacks a distinct transverse carina, by the short, declivous pronotal disc which has only hair-like setae, and by the mesocoxae separated distinctly more than the metacoxae.

*Cryphalus eriobotryae* can be distinguished from the similar *C. pruni* Eggers, 1929 by the antennal sutures which are clearly procurved, and from *C. malus* Niisima, 1909 by the smaller size (>1.7 mm vs. <1.7 mm) and by the much more stout body proportions (1.9 times as long as wide vs. 2.1 times as long as wide).

### 3.3. Female

Size 1.5 mm (paratypes 1.4–1.6 mm). Frons with weak, converging aciculations. Proportions 1.9 times as long as wide. Eye deeply emarginated. Antennae with four funicle segments (including the pedicel). Antennal club with three procurved sutures marked by dense, coarse, blunt setae. The distance between the third suture and the apex is much larger than the other sections. Pronotum dark brown, armed with six serrations. Pronotum widest at base, appearing near triangular in profile. Pronotal declivity with more than 50 asperities. Cuticular surface of pronotum with small asperities marking the anterior edge of every puncture. Pronotal disc short, distinctly sloped, approximately one fifth of the pronotal length when viewed dorsally. All setae on the dorsal face of the pronotum hair-like, and bifurcating on the hypomeron. Suture between pronotum and elytra weakly sinuate. Scutellum very small, V-shaped. Elytra 1.7 times as long as pronotum, translucent yellow-brown, broadly rounded with no clear transition to the declivity. Striae weakly visible as rows of punctures and hair-like setae. Interstrial bristles erect, weakly flattened with rounded tips. Interstrial ground vestiture tridentate, approximately twice as long as wide, with a weak iridescence. Apex of elytra obtuse. Mesocoxae moderately separated, much more than metacoxae.

### 3.4. Male

Size: 1.4–1.5 mm. Similar to females except slightly impressed frons and last ventral segment emarginated. Proventriculus typical of the genus. Sutural teeth bulbous and in two indistinct rows. Aedeagus long, weakly sclerotized. Penis apodemes almost as long as penis body.

### 3.5. Sequence Information

The sequences for cytochrome oxidase I (COI) and large nuclear ribosomal subunit (28S) were deposited in GenBank (COI: MN023143, 28S: MN023142) to enable molecular diagnosis and future phylogenetic placement. The closest matching sequence was *C. pruni* (From BOLD, COI: SCOL293-12.COI-5P, 88.1% similarity and 28S: SCOL293-12.28S-D2, 91.1% similarity).

### 3.6. Taxonomic Remarks

*Cryphalus eriobotryae* is most similar to *C. pruni* and *C. malus* morphologically and genetically. They share the simple male frons, the triangular pronotum, and the pronotal texture which has many small asperities almost to the base with a small declivous pronotal disk. They are also all known to feed on plants in Rosaceae.

### 3.7. Etymology

The name *eriobotryae* is derived from host plant genus name, *Eribotrya* Lindl., 1822. The name *Eriobotrya* is itself derived from the greek *érion* meaning woolly, and from the greek *botrys*, meaning grapes, referring to the loquat fruits.

### 3.8. Damage of Cryphalus eriobotryae in Loquat

During our investigation in Suzhou from May 2018 to May 2019, we observed that *C. eriobotryae* had infested loquat in two nonadjacent villages in Dongshan. In total, there were about 56,100 loquats trees surveyed. In the village of Caowu, with about 66 ha of loquat plantations, an estimated 90% were attacked. In Huwan, a village with about 47 ha of loquat plantations, an estimated 20% were attacked. Of the attacked trees in both Caowu and Huwan, only 5% led to mortality. In total, this pest resulted in the death of over 1000 trees. Most dead loquat trees were cultivar “Baiyu” which is the main local cultivar in the Dongshan Town.

Attack of *C. eriobotryae* on loquat is readily diagnosable initially by small (<0.8 mm) circular entrance holes (sometimes hidden in bark crevices) and frass, and later by the surrounding necrotic tissue (cankers) on the bark of the trunk and branches (Figure 2A,F). *Cryphalus eriobotryae* appears to be monogamous, with one male and one female living in an irregular chamber under the bark (Figure 2G,I). Some individual attacks on live trees seem to fail, leaving a scar on the tree (Figure 2H), but the accumulation of attackers eventually exhausts tree defenses with hundreds of brown cankers (Figure 2F). This pattern of disease is similar to thousand cankers disease, caused by a combination of a pathogenic fungus, *Geosmithia morbida* M. Kolarík, E. Freeland, C. Utley & Tisserat, 2011 (Hypocreales) and the walnut twig beetle *Pityophthorus juglandis* Blackman, 1928 [13]. No obvious signs of a fungal pathogen were observed in the diseased loquats, but the presence of pathogens was not systematically investigated. Successful reproduction can be detected by the presence of many larvae, pupae and adults under the bark (Figure 2B–E). All life stages of *C. eriobotryae* were observed within galleries in the field. Large numbers of small, circular exit holes were noted after the trees have wilted. These holes were excavated by the emerging new generation of beetles. Infestations occurred on the basal part of trunk at first, then expanded upward. Attacks were concentrated on the trunk but also occurred on branches with minimum diameter of approximately 1 cm. The scars of cankers from failed attacks were present on some living trees (Figure 2H). Additional symptoms of infestation included the retention of dead leaves on the infested trees (Figure 2A).

During our field investigations we found less mortality on the loquat cultivar “Hongyu”, which grows faster than “Baiyu” and is usually selected as rootstock in grafting at local plantation. This is the first published record of epidemic mortality of loquat in China known to us.

Given the distribution of the plant genus *Eriobotrya* and the distribution of similar *Cryphalus* species, we suspect that the beetle is native to East or Southeast Asia. According to local farmers, rootstocks were imported from other regions of China in 2017, which could have been a route of local introduction.

It is unclear why the pest has not been reported before from Suzhou or other regions, and may represent unreported changes in environmental conditions or management techniques. If the beetle was recently introduced to Suzhou, the existing management techniques may be naïve to the presence of the beetle.

The threat to other Asian loquat varieties and species is unknown. There are at least 21 species of *Eriobotrya* plants which are native to China [14]. This beetle could be a pest to all Chinese or Asian loquat plantations.

There are 1100 ha of loquat plantations in the Dongshan Town with an annual value of production of 24 million USD [6]. The economic loss of *C. eriobotryae* is difficult to estimate because it would take more than 10 years between initial planting and harvest fruit production. At this time, we have no options for effective management of *C. eriobotryae*.

## Figures and Tables

**Figure 1 insects-10-00180-f001:**
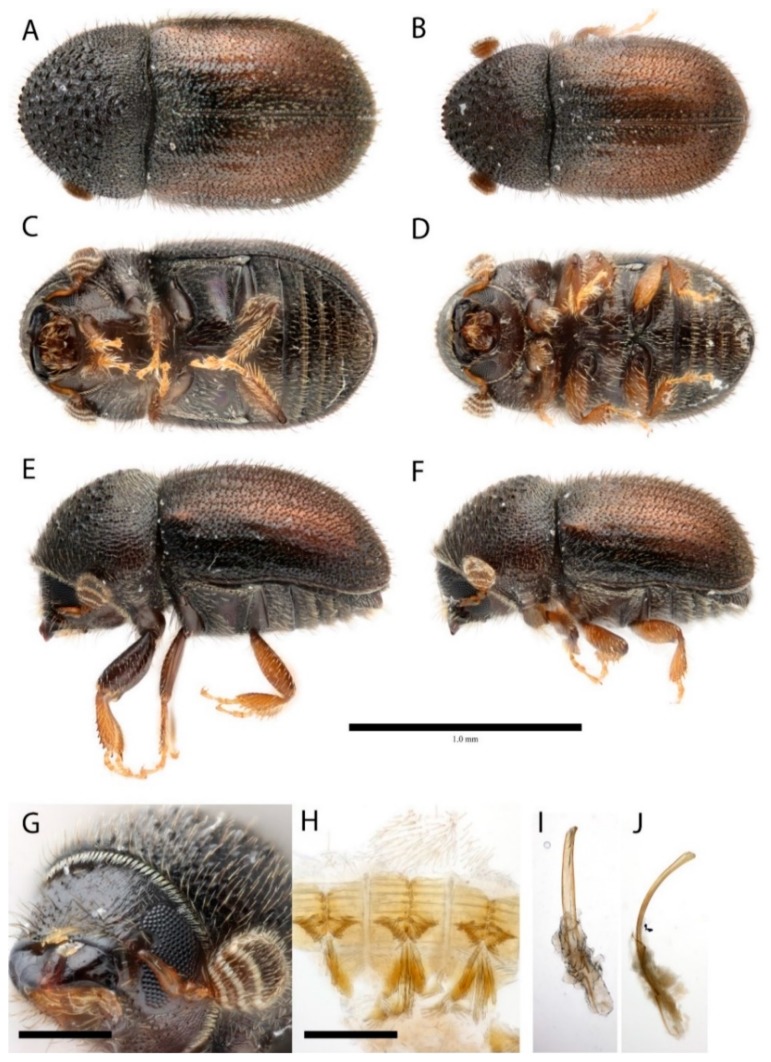
Images of *Cryphalus eriobotryae*. (**A**,**B**) Dorsal view; (**C**,**D**) ventral view; (**E**,**F**) lateral view; (**G**) head; (**H**) proventriculus; (**I**,**J**) male genitalia; (**A**,**C**,**E**,**G**) holotype (female); (**B**,**D**,**F**,**H**,**I**,**J**) paratype (male). Images courtesy of Andrew Johnson, UF Forest Entomology Collection, (CC BY 3.0).

**Figure 2 insects-10-00180-f002:**
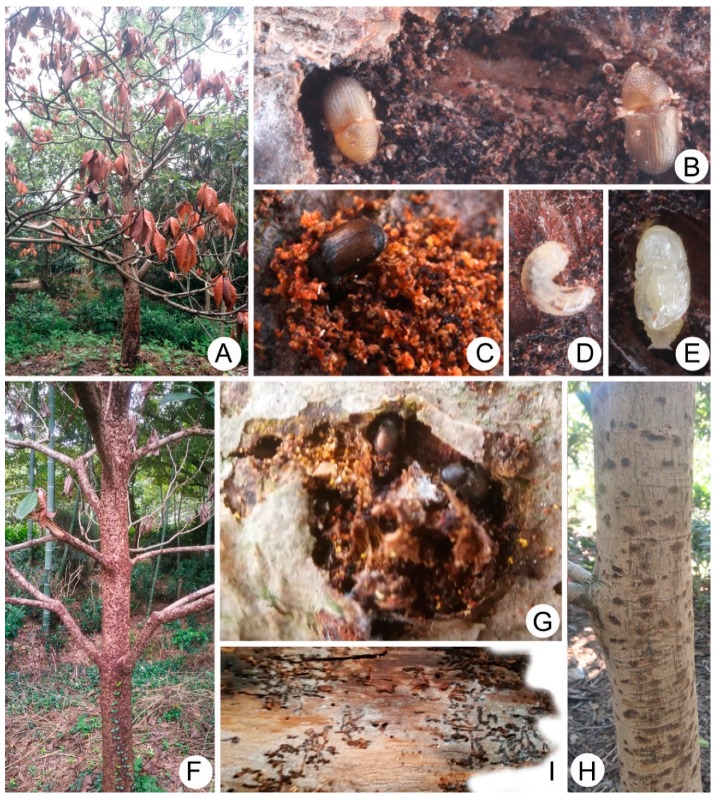
(**A**) Tree killed by *Cryphalus eriobotryae*; (**B**) teneral adult; (**C**) adult outside a newly excavated gallery; (**D**) larva; (**E**) pupa; (**F**) cankers on the trunk; (**G**) adults made the canker; (**H**) tree recovered after failed *C. eriobotryae* attack; (**I**) galleries on the underside of peeled bark.

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
