# Peer review of "Cryphalus eriobotryae sp. nov. (Coleoptera: Curculionidae: Scolytinae), a New Insect Pest of Loquat Eriobotrya japonica in China"

_insects, 2019, doi:10.3390/insects10060180_

Round 1

Reviewer 1 Report

Scientific name should be written in Italic.

I think the paragraph and Figurers number don't match each others.

In my opinion, Line 147 - Figure 2E -> 2F, Line 149 - Figure 2F -> 2G, Line 151 - Figure 2G ->2F maybe better.

Author Response

Response to Reviewer 1 Comments

Thank you for your review. All changes in the manuscript were red font color this time.

1-1.  All the scientific names were written in Italic

Response 1-1: All the scientific names were written in Italic

1-2.  I think the paragraph and Figurers number don't match each others.

In my opinion, Line 147 - Figure 2E -> 2F, Line 149 - Figure 2F -> 2G, Line 151 - Figure 2G ->2F maybe better.

Response 1-2: We have updated the figure number.

Reviewer 2 Report

The work presented, as a whole, is cohesive, and written in a clear manner, except for the small changes made, which are attached to the pdf. The argument concerns the discovery of a new species and altogether, is interesting and scientifically accurate. An exception is the inaccurate word choice of "canker", as very often the Scolytidae have species-specific tunnels; therefore, the use of the word "canker" should be reconsidered. As well, it is essential that the tunnel system, in terms of shape and dimension, is described accurately.
The suggestion to add additional photos or detailed drawings of the tunnels would be appropriate as it would provide all the elements useful for immediate recognition of the new species, even on new and different hosts from the loquat. The journal also suggests that there are no elements that rule out the possible presence of associated fungi.
The bibliography consulted was found to be correct and precise. No plagiarism was detected. English does not require further modifications except those shown in the attached pdf.
For the benefit of describing tunnel systems, i advise this article: Wainhouse, D., Murphy, S., Greig, B., Webber, J., & Vielle, M. (1998). The role of the bark beetle Cryphalus trypanus in the transmission of the vascular wilt pathogen of takamaka (Calophyllum inophyllum) in Seychelles. Forest Ecology and Management, 108(3), 193-199.

Author Response

Response to Reviewer2 Comments

Thank you for your review. We accepted all the suggestions in manuscript and highlight them with red font color.

1.     Line 15, change “Our survey of plantation records” to “The survey of plantation records”.

2.     Line 17, change” showed signs of attack” to” showed signs of infestation”.

3.     Line 18, change “Mass attacks of the apparently loquat-specific” to “Outbreaks of the apparently loquat-specific”.

4.     Line 20, change “This pest is of concern as plantation pest…” to “This pest is of concern as loquat plantation pest ….”.

5.     Line 43, change “several reported loquat trees” to “some loquat trees”.

6.     Line 60, change “stereo microscope” to “stereomicroscope”

7.     Line 77, change” DNEasy” to “DNeasy”.

8.     Line 146, “Defining them as "cankers" may not be correct. Furthermore, there is no description of the tunnel system and of the feeding / brooding geometry.”

Response: We understand ‘canker’ is a term commonly used for plant diseases representing regions of necrotic tissue as seen following attack of this beetle. We reworded a sentence for clarity. The damage is similar to that of thousand canker disease on walnut in USA. For the tunnel system, we did not systematically investigate the shape of the gallery, but this if often an irregular cave, and not informative for Cryphalus as in other bark beetles. We added a photo of older galleries to show the approximate structures of galleries. (Figure 2I).

9.  No obvious signs of fungal pathogen were observed in the 153 diseased loquats, but this was not systematically investigated.

   “If they have not been investigated, how can you say that they are not present?”

Response: We use the word “signs” from a plant pathology perspective to indicate visible fungal structures such as fruiting bodies or fungus visible in the gallery. From the tunnel and gallery photo and our field observation, we did not found any fungi-like structures in the gallery or chamber, though other methods which we did not use may reveal that there are fungi there. The phrase “systematically investigated” refers to attempts to characterize, identify or isolate fungi in the tissue, which we did not do. We did use the term “signs” later in the manuscript incorrectly and have updated this (line 159, changed from “signs” to “symptoms”)

      10. Line 173  naïve to the presence of 173 the beetle.

   “disregard” naïve

Response: The wording before this was misleading. We have edited the sentence, still including the word ‘naïve’, emphasize that the beetle may have been locally introduced and being a pest there only due to different environmental conditions or management techniques.

      11. For the benefit of describing tunnel systems, i advise this article: Wainhouse, D., Murphy, S., Greig, B., Webber, J., & Vielle, M. (1998). The role of the bark beetle Cryphalus trypanus in the transmission of the vascular wilt pathogen of takamaka (Calophyllum inophyllum) in Seychelles. Forest Ecology and Management, 108(3), 193-199

Response: The paper suggested does not accurately describe the gallery system of Cryphalus spp. in general, and we did not study whether some of the galleries appeared to be non-reproductive galleries into pith of leaf scares as described in the paper. Cryphalus galleries in general are interesting because the larvae feed communally for the first instar then usually (but not always) make individual tunnels as they get larger. This was not investigated for the species on loquat and hopefully this paper will facilitate further study.

Reviewer 3 Report

I read carefully and with interest the manuscript “Cryphalus eriobotryae sp. nov. (Coleoptera: Curculionidae: Scolytinae), a new insect pest of loquat Eriobotrya japonica in China” submitted for publication in Insects journal. The authors described a new Cryphalus species, wich was developed on Eriobotrya japonica. The authors describe numerous morphological details that distinguish this species from other Cryphalus species. The genetic analyses confirms this assumption. The atack characteristics of this species are very good described. It is interesting that this is the first report  of Cryphalus eriobotryae local outbreak. 

Certainly new research is needed on the origin and spread of this species.

The figures (images) are very sugestive (very good quality).

In this context, in my opinion, it deserves publication in the Insects Journal.

Author Response

Response to Reviewer 3 Comments

Thank you for your review. We are glad you like our manuscript and figures. Yes, we agree more researches are needed on the origin and spread of this species. That’s why we wrote this paper so that more people could pay attention to this pest.